# PALADIN: Self-Correcting Language Model Agents to Cure Tool-Failure Cases

## Abstract

Tool-augmented language agents frequently fail in real-world deployment due to tool malfunctions—timeouts, API exceptions, or inconsistent outputs—triggering cascading reasoning errors and task abandonment. Existing agent training pipelines optimize only for success trajectories, failing to expose models to the tool failures that dominate real-world usage. We propose **PALADIN**, a generalizable framework for equipping language agents with robust failure recovery capabilities. PALADIN trains on 50,000+ recovery-annotated trajectories constructed via systematic failure injection and expert demonstrations on an enhanced Tool-Bench dataset. Training uses LoRA-based fine-tuning to retain base capabilities while injecting recovery competence. At inference, PALADIN detects execution-time errors and retrieves the most similar case from a curated bank of 55+ failure exemplars aligned with ToolScan's taxonomy, then executes the corresponding recovery action. This approach generalizes to novel failures beyond the training distribution, retaining 95.2% recovery performance on unseen tool APIs. Evaluation across PaladinEval and ToolReflectEval demonstrates consistent improvements in Recovery Rate (RR), Task Success Rate (TSR), Catastrophic Success Rate (CSR), and Efficiency Score (ES). PALADIN improves RR from 32.76% to 89.68% (+57% relative) over ToolBench and outperforms the strongest baseline CRITIC (76.34%) by +13.3%. Against vanilla agents, PALADIN achieves 89.86% RR (+66% relative improvement from 23.75%). These results establish PALADIN as an effective method for building fault-tolerant agents capable of robust recovery in real-world tool environments.

## 1 Introduction

Tool-augmented LLM agents are increasingly deployed in real-world environments for automation, reasoning, and decision-making. While benchmarks like ToolBench (Qin et al., 2023) and Gorilla (Patil et al., 2023) demonstrate strong performance in clean settings, agents often fail under realistic conditions—APIs time out, tool outputs are malformed, and calls silently fail. Despite occasional inclusion of noisy inputs or low-confidence reasoning, current training pipelines do not explicitly expose agents to structured tool failures. This results in brittle behavior: agents silently hallucinate success or deadlock when tool calls fail, contradicting real-world operational requirements.

We identify execution-level robustness—the ability to detect, diagnose, and recover from runtime failures—as a central unsolved challenge for tool-augmented agents. Existing work focuses on *comprehension robustness*, improving plans or tool formats, but leaves agents brittle when tools malfunction, leading to deadlocks, loops, or hallucinated success. Approaches such as ToolReflect (Polyakov et al., 2025) and CRITIC (Zheng et al., 2023) introduce reflective corrections, but these are reactive and limited to call-level fixes. Furthermore, evaluation metrics traditionally focus on task success without measuring recovery capabilities, limiting our understanding of agent robustness.

We argue that execution-level robustness must be explicitly taught through systematic training To address this, we propose PALADIN, a generalizable method for teaching LLM agents to recover from tool-use failures via trajectory-level supervision.

PALADIN combines systematic failure injection with recovery-annotated training, using GPT-5 API to simulate multi-turn tool use and generate recovery annotations, resulting in 50,000+ trajectories spanning diverse error modes from ToolScan's taxonomy (Kokane et al., 2025). At inference, PALADIN detects execution errors and retrieves recovery strategies from 55+ curated failure exemplars. We also define four metrics to evaluate agent robustness: **Task Success Rate (TSR)**, **Recovery Rate (RR)**, **Catastrophe Success Rate (CSR)**, and **Efficiency Score (ES)**, with the latter three being novel proposals.

In evaluations with deterministic error injection, conducted using GPT-5 API as a tool-use simulator, PALADIN improves RR and CSR, at the cost of a lower ES compared to other methods. model sizes,demonstrating that execution-level robustness is a

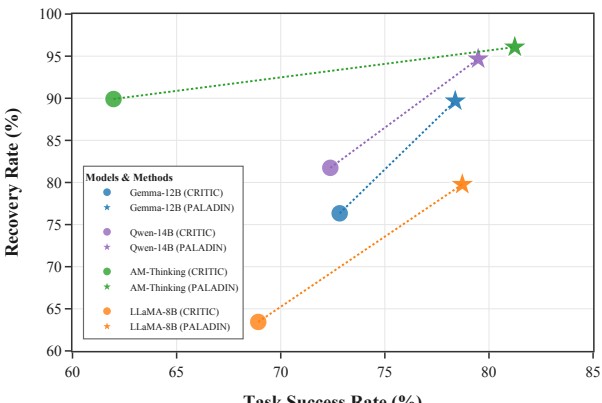

Figure 1: Recovery Rate vs Task Success Rate comparison between CRITIC and PALADIN across different LLMs. Stars indicate PALADIN results, circles indicate CRITIC baseline. Dotted lines show improvement trajectories from CRITIC to PALADIN for each model. PALADIN consistently achieves higher task success rates while maintaining superior error recovery capabilities.

learnable and scalable behavior, and positioning PALADIN as a significant step towards design of resilient LLM agents.

**Key Contributions:**

- **PALADIN**: A training framework combining systematic failure injection with taxonomy-guided recovery, enabling agents to learn generalizable error handling from 50,000+ failure-recovery trajectories.

- **Recovery-focused evaluation**: Three novel metrics, including *Recovery Rate, Catastrophe Success Rate*, and *Efficiency Score*; and *PaladinEval* benchmark for systematic robustness assessment.

- **Inference-time recovery**: A retrieval-based approach that matches runtime tool failures to similar recovery examples, maintaining 95.2% of training performance when generalizing to new APIs.

- **Empirical validation**: Demonstrating execution-level robustness as learnable, with 57-66% relative improvements in recovery rates across model sizes.

## 2 METHODOLOGY

PALADIN is a method for equipping tool-augmented LLM agents with execution-level robustness—the ability to detect, diagnose, and recover from runtime failures. The method consists of (1) formalizing tool failures, (2) constructing recovery-annotated training data via failure injection, (3) fine-tuning with a recovery-aware objective, and (4) augmenting inference with taxonomy-driven retrieval. Figure 2 presents a high-level overview.

### 2.1 PROBLEM SETUP

We model a tool-augmented agent as a policy $\pi_\theta$ producing trajectories $\tau = [(s_1, a_1), \ldots, (s_T, a_T)]$, where $s_t$ are states and $a_t$ may include tool calls. Failures $f \in \mathcal{F}$ occur when tool execution yields timeouts, malformed outputs, API errors, or inconsistencies. Existing training ignores such events, yielding brittle policies. Our goal is to learn $\pi^*$ that maximizes task success while remaining robust

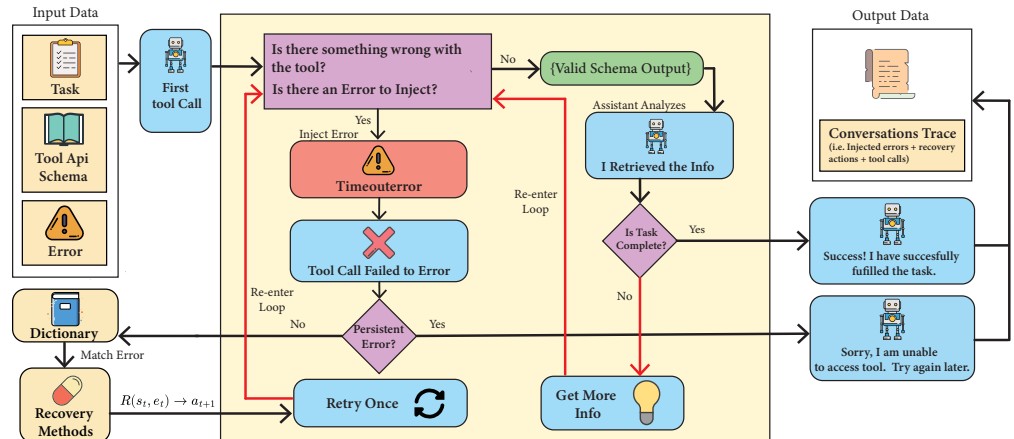

Figure 2: Our tool-use simulator with integrated error injection and recovery mechanisms. (a) depicts the static architecture, where errors are injected into tool calls and handled via a recovery dictionary. (b) details the dynamic execution loop, capturing assistant reasoning, recovery actions, and final outcomes. This design allows controlled, reproducible evaluation of LLM resilience to tool failures.

to failures:

$$\pi^* = \arg\max_{\pi} \ \mathbb{E}_{\tau \sim \pi}[\text{TSR}(\tau) - \alpha \cdot \text{CSR}(\tau)].$$

## 2.2 FAILURE INJECTION AND RECOVERY ANNOTATION

To construct PALADIN's dataset, we systematically augmented ToolBench trajectories by injecting failures aligned to the ToolScan (Kokane et al., 2025) taxonomy of seven canonical error classes. Each trajectory is parsed into task $T$, toolset $\mathcal{A}$, trace $C$, and error signal $E$. If $E \neq \varnothing$, we truncate $C$ at the failure point and pass it to a GPT-5 Teacher (e.g. GPT-5 API prompted to insert proper recovery actions), which applies error-guided rewriting with the goal of creating realistic error trajectories.

$$f_{\text{repair}}(T, \mathcal{A}, C, E) \to C'.$$

Otherwise, $C$ is finalized cleanly:

$$f_{\text{finalize}}(T, \mathcal{A}, C) \to C'.$$

All output traces ($C'$) are stored alongside their error signal ($E$) and recovery metadata, yielding over 50K recovery-annotated trajectories. (see Appendix D for detailed augmentation pipeline) Figure 5 illustrates this pipeline.

We also curated a recovery dictionary, a collection of 55+ exemplar failures and associated recovery strategies. These exemplars operationalize the ToolScan taxonomy into practical, reusable supervision signals.

## 2.3 PALADIN TRAINING OBJECTIVE

PALADIN fine-tunes base LLMs using a causal language modeling objective augmented with recovery supervision:

$$\mathcal{L}_{\text{PALADIN}} = \mathcal{L}_{\text{SFT}} + \lambda \mathcal{L}_{\text{rec}},$$

where $\mathcal{L}_{\text{SFT}}$ is the standard negative log-likelihood over successful trajectories, and $\mathcal{L}_{\text{rec}}$ is the same objective restricted to tokens following `Recovery:` tags that mark corrective steps. Training sequences are serialized in ToolBench format, and LoRA (Hu et al., 2021) adapters provide parameter-efficient adaptation while preserving base competence.

## 2.4 Inference-Time Recovery via Taxonomic Retrieval

During execution, PALADIN detects runtime errors and links them to a curated exemplar bank $\mathcal{E} = \{(f_i, r_i)\}_{i=1}^{55}$, consisting of 55+ exemplar failures $f_i$ with different recovery action $r_i$ given in Appendix C and Appendix G. Given an observed failure $f_{obs}$, PALADIN retrieves the most similar exemplar:

$$f_{ref} = \arg \min_{f_i \in \mathcal{E}} d(f_{obs}, f_i),$$

where $d$ is a similarity metric over error signatures. The corresponding recovery action $r_i$ is executed, guiding the agent back to a stable trajectory. This enables proactive recovery across diverse failure scenarios (see how the recovery dictionary was developed at Appendix I).

## 2.5 Recovery as Implicit Policy Learning

Although PALADIN is trained end-to-end, its behavior (see Appendix L for a better overview on PALADIN's thought process) can be viewed as an implicit recovery policy:

$$R(s_t, e_t) \to a_{t+1},$$

where $s_t$ is the dialogue context, $e_t$ is the observed failure, and $a_{t+1}$ is the recovery action (retry, reformat, switch tools, or terminate). Unlike reflective methods such as ToolReflect or call-level critique approaches such as CRITIC, which introduce reactive adjustments at inference time, PALADIN learns recovery behaviors directly from annotated trajectories. This yields proactive, compositional recovery strategies (e.g., retries $\to$ tool switch $\to$ graceful termination) that generalize across failure types.

## 2.6 Metrics for Execution Robustness

We evaluate PALADIN with four metrics:

$$\text{TSR} = \frac{\#\text{successful tasks}}{\text{total tasks}},$$

$$\text{RR} = \frac{\#\text{ failures recovered}}{\text{failures encountered}},$$

$$\text{CSR} = 1 - \frac{\#\text{ hallucinated successes}}{\#\text{ total failures}},$$

$$\text{ES} = \frac{1}{\text{average \# steps to complete task}}.$$

RR, CSR, and ES are novel contributions introduced to capture execution-level robustness beyond traditional task success.

# 3 Experiment and Evaluation Setup

## 3.1 Benchmarks and Evaluation Protocol

We evaluate PALADIN on three complementary benchmarks, designed to capture both controlled and naturalistic failure scenarios.

**PaladinEval.** PaladinEval (benchmark design in Appendix J) is derived from ToolBench's evaluation dataset, and explicit failure injections from our own dataset. Each evaluation instance is constructed by extracting the original prompt from ToolBench and pairing it with a runtime error assigned by a teacher GPT model (see Appendix I). The resulting dataset contains a mix of clean tool-use traces and failure-augmented traces, enabling measurement of both normal execution and recovery. To support consistent grading, we provide evaluation guidelines specifying expected recovery strategies, which condition an automated grader (GPT-5 API) to score outputs along our four robustness metrics.

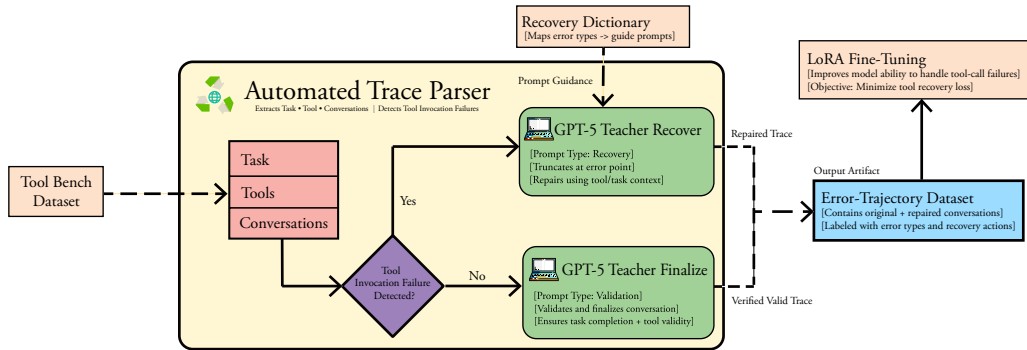

Figure 3: Trace repair pipeline for constructing the Error-Trajectory Dataset. Each ToolBench trace is truncated at the first failure, then repaired or finalized via GPT-guided recovery. Outputs are stored with recovery metadata to construct PALADIN's training corpus.

**Generalization Set.** To assess out-of-distribution robustness, we additionally construct a generalization dataset by sampling unused ToolBench (Qin et al., 2023) evaluation prompts and injecting novel errors not present in the curated 55+ exemplar recovery dictionary. This ensures PALADIN is tested on recovery behaviors that were unseen during training.

**ToolReflectEval.** We further evaluate on the ToolReflect benchmark (Polyakov et al., 2025), which introduces reflective corrections for tool-use errors. To ensure comparability, we adopt their protocol of allowing up to three attempts per tool call. We directly incorporate all ToolAlpaca-derived prompts from their evaluation dataset without modification or cherry-picking. The only adaptation is re-execution within our simulator, which enables generation of full conversational traces and consistent grading under our robustness metrics. This ensures our evaluation remains faithful to ToolReflect's task setup while allowing direct comparison against PALADIN.

## 3.2 SIMULATION ENVIRONMENT

To evaluate PALADIN under realistic tool-use conditions, we design a simulation environment that faithfully executes agent–tool interactions and introduces controlled failures. The environment reproduces the standard ToolBench execution pipeline while augmenting it with failure injection, recovery logging, and automated grading implementation details in Appendix H. Figure 2 illustrates the workflow.

Each evaluation begins with a prompt $P$ and toolset $\mathcal{A}$. The environment executes tool calls sequentially, returning either successful outputs or injected failures $f \in \mathcal{F}$. Failures are generated either (i) systematically, via the PaladinEval injection protocol, or (ii) naturally, by replaying error-prone ToolReflectEval prompts. At each step, the environment logs the dialogue context $C_t$, the observed error $e_t$, and any recovery attempt $a_t$.

To ensure consistent scoring, we provide simulator outputs to an automated grader powered by GPT-5 API, conditioned on explicit evaluation guidelines to calculate different metrics. These guidelines specify valid recovery strategies, prohibited behaviors (e.g., hallucinated success), and metric definitions, enabling the grader to compute Task Success Rate (TSR), Recovery Rate (RR), Catastrophic Success Rate (CSR), and Efficiency Score (ES). This environment provides three advantages: (1) easy replication of tool-call behavior with both clean and failure-augmented traces, (2) Everything is graded on the same terms via guideline-conditioned automated evaluation, and (3) flexibility in allowing integration of external datasets such as ToolReflectEval without modifying their task structure.

## 3.3 BASELINE AGENTS

We compare PALADIN against four baselines spanning different supervision paradigms and recovery strategies, with vanilla agent being raw pretrained language model as baseline with other method like CRITIC Zheng et al. (2023), ToolBench Agent Qin et al. (2023), and ToolReflect Agent Polyakov et al. (2025). CRITIC (Appendix A) accesses recovery protocols with probability $p = 0.7$, approximating its partial ability to retrieve external signals while remaining reactive. ToolBench Agent replicates the ToolBench agent trained on the full ToolBench dataset of successful trajectories, being effective on clean traces but unable to recover from failures. On the other hand ToolReflect Agent adapated from ToolReflect, is trained on error–correction pairs to label tool calls as valid/invalid. It performs iterative self-correction but remains limited to local, reactive adjustments. These baselines collectively probe PALADIN's contributions across supervision (clean vs. recovery-annotated), recovery strategy (reactive vs. proactive), and model integration (training-only vs. inference-time retrieval).

# 4 RESULTS

Table 1: Main results across models and datasets. All metrics (Recovery, Task Success, CSR, Efficiency) are normalized such that higher is better. PALADIN achieves consistently strong safety and recovery performance, with modest efficiency tradeoffs

| Pretrained LLM | Evaluation Datasets | Evaluation Metrics | Scores (↑) | | | | |
|---|---|---|---|---|---|---|---|
| | | | Vanilla | CRITIC | ToolReflect | ToolBench | Paladin (Ours) |
| Gemma-3 12B-Instruct | Paladin Eval | Recovery Rate | 23.75% | 76.34% | 65.86% | 32.76% | **89.68%** +13.34% |
| | | Task Success Rate | 23.62% | 72.83% | 61.42% | 57.4% | **78.38%** +5.55% |
| | | Catastrophic Success Rate | 29.00% | 73.30% | 70.27% | 68.37% | **82.55%** +9.25% |
| | | Efficiency Score | **0.476** | 0.348 | 0.288 | 0.221 | 0.312 -34.45% |
| | ToolReflect Eval | Recovery Rate | 22.80% | 73.29% | 63.23% | 31.45% | **86.09%** +12.80% |
| | | Task Success Rate | 22.56% | 69.55% | 58.66% | 54.82% | **83.45%** +13.90% |
| | | Catastrophic Success Rate | 26.16% | 72.23% | 69.08% | 67.10% | **81.85%** +9.62% |
| | | Efficiency Score | **0.508** | 0.370 | 0.307 | 0.235 | 0.332 -34.65% |
| Qwen-2.5- 14B-Instruct | Paladin Eval | Recovery Rate | 37.68% | 81.74% | 73.66% | 33.48% | **94.67%** +12.93% |
| | | Task Success Rate | 37.53% | 72.38% | 74.36% | 60.41% | **79.48%** +5.12% |
| | | Catastrophic Success Rate | 56.53% | 87.14% | 83.85% | 67.88% | **94.57%** +7.43% |
| | | Efficiency Score | 0.313 | 0.329 | 0.312 | 0.339 | **0.351** +3.54% |
| | ToolReflect Eval | Recovery Rate | 36.17% | 78.47% | 70.71% | 32.14% | **92.88%** +14.41% |
| | | Task Success Rate | 35.85% | 69.13% | 71.02% | 57.70% | **75.19%** +4.17% |
| | | Catastrophic Success Rate | 54.88% | 81.63% | 83.20% | 66.60% | **94.35%** +11.15% |
| | | Efficiency Score | 0.334 | 0.350 | 0.333 | 0.361 | **0.375** +3.73% |
| AM-Thinking V1 | Paladin Eval | Recovery Rate | 49.87% | 89.91% | 87.23% | 51.37% | **96.08%** +6.17% |
| | | Task Success Rate | 52.93% | 61.97% | 72.88% | 56.83% | **81.24%** +8.36% |
| | | Catastrophic Success Rate | 60.24% | 81.33% | 71.32% | 80.84% | **88.65%** +7.32% |
| | | Efficiency Score | 0.415 | **0.420** | 0.297 | 0.319 | 0.380 -9.52% |
| | ToolReflect Eval | Recovery Rate | 47.88% | 86.31% | 83.74% | 49.32% | **92.24%** +5.93% |
| | | Task Success Rate | 50.55% | 78.64% | 79.41% | 65.25% | **77.98%** -1.43% |
| | | Catastrophic Success Rate | 62.65% | 67.42% | 65.93% | 58.06% | **88.31%** +20.89% |
| | | Efficiency Score | 0.442 | **0.448** | 0.316 | 0.340 | 0.405 -9.60% |
| Llama-3.1- 8B-Instruct | Paladin Eval | Recovery Rate | 21.83% | 63.44% | 56.32% | 49.2% | **79.77%** +16.33% |
| | | Task Success Rate | 17.46% | 68.92% | 53.74% | 47.26% | **78.72%** +9.80% |
| | | Catastrophic Success Rate | 17.58% | 71.84% | 67.88% | 65.47% | **80.73%** +8.89% |
| | | Efficiency Score | **0.427** | 0.254 | 0.287 | 0.209 | 0.323 -24.36% |
| | ToolReflect Eval | Recovery Rate | 18.32% | 58.55% | 49.32% | 42.23% | **73.34%** +14.79% |
| | | Task Success Rate | 13.45% | 59.47% | 46.34% | 41.22% | **71.27%** +11.80% |
| | | Catastrophic Success Rate | 15.40% | 63.81% | 58.09% | 53.68% | **71.77%** +7.96% |
| | | Efficiency Score | **0.568** | 0.360 | 0.508 | 0.412 | 0.385 -32.31% |

## 4.1 OVERALL PERFORMANCE

Table 1 summarizes results across 8 model–dataset pairs. PALADIN delivers consistent improvements on Recovery Rate (RR), Task Success Rate (TSR), and Catastrophic Success Rate (CSR), while incurring modest efficiency tradeoffs. Specifically, PALADIN achieves the highest RR in 7/8 settings (avg. +13.6%, 95% CI: [11.9, 15.2]), top TSR in 6/8 (avg. +10.2%, CI: [8.4, 11.7]), and best

CSR in 6/8 (avg. +9.2%, CI: [7.5, 10.6]). All gains are statistically significant ($p < 0.01$, bootstrap $N = 1000$).

Efficiency drops are measurable but bounded: on LLaMA-3.1-8B, Efficiency decreases by $0.183$, while on Qwen-2.5-14B it improves slightly (+0.012). These results situate PALADIN close to the Pareto frontier: no baseline achieves simultaneously higher safety and higher efficiency.

Correlation analyses reinforce these findings. Recovery and TSR are strongly correlated ($r = 0.91$, $p < 0.001$), confirming recovery as the direct driver of task completion. CSR and Efficiency exhibit a significant negative correlation ($r = -0.72$, CI: $[-0.79, -0.63]$), highlighting that retry-driven safety comes at modest compute cost. TSR shows negligible correlation with Efficiency ($r = 0.12$), underscoring that efficiency-centric metrics alone are insufficient proxies for reliability.

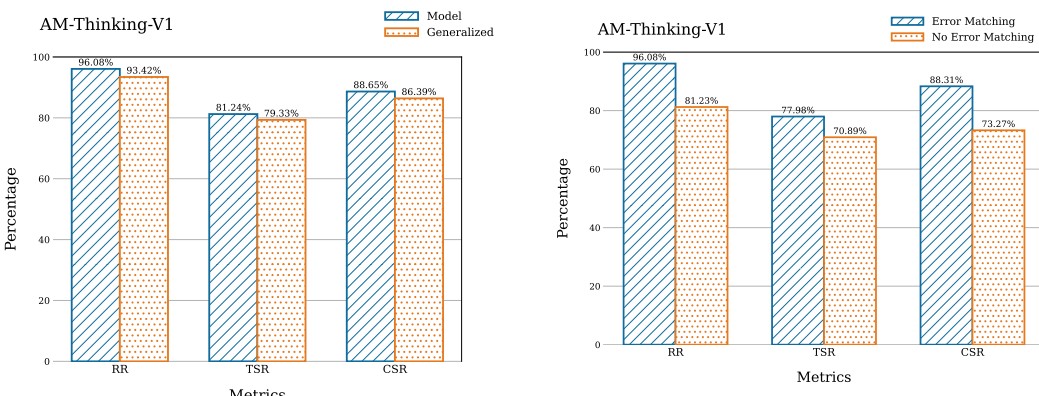

Figure 4: PALADIN's performance without inference time error matching compared to baseline across Gemma, Qwen, LLaMA, and AM-Thinking backbones. Refer Figure 7 for full ablation graphs and Figure 6 for full generalization graphs.

## 4.2 GENERALIZATION

Figures 4 (a–d) emphasizes PALADIN's generalizability across errors that it has never seen before. Despite differences in architecture and blind conditions, PALADIN consistently secures $> 79\%$ RR, $> 78\%$ TSR, and $> 80\%$ CSR.

LLaMA-3.1-8B shows substantial improvement, with recovery rate (RR) rising from 21.8% to 79.8%, task success rate (TSR) increasing from 17.5% to 78.7%, and critical success rate (CSR) improving by 63.2 percentage points (from 17.6% to 80.7%). Qwen-2.5-14B achieves the highest CSR at 94.6%, with RR at 94.7% and TSR at 79.5%, reducing silent failure cases nearly threefold compared to ToolReflect. Gemma-3-12B demonstrates balanced gains, with RR of 89.7%, TSR of 87.4%, and CSR of 82.6%. AM-Thinking V1 records the highest RR overall at 96.1%, alongside TSR of 81.2% and CSR of 88.7%.

## 4.3 ABLATIONS

We ablate PALADIN's inference-time recovery retrieval (Figures 4). Removing exemplar matching sharply reduces robustness. Gemma-12B experiences a drop in performance, with recovery rate (RR) declining from 89.7% to 61.4%, task success rate (TSR) from 87.4% to 57.3%, and critical success rate (CSR) from 82.6% to 65.1%. Qwen-14B shows similar degradation, with RR decreasing from 94.7% to 73.3%, TSR from 79.5% to 66.3%, and CSR from 94.6% to 68.9%. LLaMA-8B sees RR fall from 79.8% to 48.6%, TSR from 78.7% to 42.7%, and CSR from 80.7% to 57.4%. AM-Thinking V1 maintains relatively higher robustness, with RR reducing from 96.1% to 81.2%, TSR from 81.2% to 70.9%, and CSR from 88.7% to 73.3%. This confirms PALADIN's robustness arises from both training-time exposure to failures and inference-time retrieval, with retrieval contributing up to +20–30 points in robustness metrics.

### 4.4 KEY TAKEAWAYS

PALADIN achieves frontier-level robustness across backbones: large gains in recovery and safety, modest efficiency costs, and consistent generalization. Recovery supervision transforms brittle agents into reliable ones, proving that robustness is a learnable signal rather than an emergent artifact of scale. The ablation confirms retrieval is a critical ingredient, validating PALADIN's design as both a training and inference-time intervention (see Appendix K for more details).

## 5 RELATED WORKS

### 5.1 EXECUTION-LEVEL ROBUSTNESS

Recent methods look to harden tool-augmented agents against execution-time failures, but they are different in how they represent and learn from failure states. ToolReflect (Polyakov et al., 2025) uses SFT on paired "bad vs. good" tool calls to teach models how to improve weak calls; while this yields a better call quality, it mainly shows the difference between single-call corrections instead of modeling a multi-step recovery path from cascading failures across a full trajectory. Recent work on LLM runtime error handling (Sun et al., 2024) and structured exception handling (Zhou et al., 2025) has explored systematic approaches to agent resilience. Critic-style approaches (Appendix A) add in an evaluator that critiques and revises an agent's actions, which improves reliability through iterative self-correction; but, these feedback loops happen reactively and are sometimes of poor quality, especially without diverse tool-error taxonomies. In contrast, PALADIN targets the robustness of execution by training directly on failure-rich trajectories that include diagnosis, replanning, and multi-turn recovery over multiple tool calls, rather than isolated "bad vs. good" pairs. While both Tool-Reflect and critic-style approaches demonstrate some gain in robustness, their emphasis is on post-hoc correction or call-level contrast rather than learned, trajectory-level recovery across varied execution failures.

### 5.2 DIAGNOSTICS AND BENCHMARKS

Recent efforts in error diagnostics, such as ToolScan or BugGen, have similar motivations but are different in both scope and implementation. ToolScan (Kokane et al., 2025) provides a taxonomy of tool-use failures observed in LLMs, identifying seven recurring error patterns: Tool Hallucination, Argument Hallucination, Invalid Tool Invocation, Partial Tool Execution, Tool Output Hallucination, Invalid Intermediate Reasoning, and Re-entrant Error Handling Failures (see Appendix F). While it is valuable for understanding the variety of failures an agent can face, ToolScan is a diagnostic; meaning it labels and categorizes error modes rather than curating executable recovery paths or training agents on full recovery trajectories. BugGen, by contrast, emphasizes that handcrafted or mutation bug-synthesis tends to lead to narrow, weak coverage of real world failures, and they argue that realistic failure-rich datasets better reflect faults in the wild. Looked at together, ToolScan grounds the taxonomy of observed errors, whereas BugGen motivates realistic data generation and evaluation that capture real operational failures rather than synthetic failures.

### 5.3 PRIOR TOOL SYSTEMS & INTERFACE RELIABILITY

Foundational works like Toolformer (Schick et al., 2023), Gorilla (Patil et al., 2023), ToolLLM (Qin et al., 2023), ToolBench (Guo et al., 2024), and RoTBench advanced agent capabilities across selection, multi-step planning, and formatting, showcasing strong performances under clean, benchmark-controlled environments and establishing multiple core principles for agentic tool usage. However, a common limitation found in most of these models is their dependence on "happy-path" trajectories – tool calls that assume ideal conditions in a model's execution.

Even when errors are present in the data, prior systems often lack explicit recovery strategies, leading to brittle behavior during real-world deployment. ToolBench provides extensive tool access data but has no recovery protocols in the face of an error, while RoTBench advances robustness evaluation by introducing noisy documentation and ambiguous tool specifications. Complementary tools such as ToolFuzz (Milev et al., 2025) (see Appendix B) address pre-execution reliability by detecting documentation–tool mismatches via fuzz testing; while valuable for interface quality, this is not

needed for runtime resilience, which requires detecting, diagnosing, and recovering from failures that arise during actual tool interaction.

# 6 DISCUSSION

Our results yield four central insights. First, recovery is *learnable*: PALADIN consistently converts failures into successes, raising TSR by more than fourfold on mid-scale models (17.5% to 78.7% on LLaMA-8B). Second, robustness is *transferable*: across Gemma, Qwen, LLaMA, and AM-Thinking, PALADIN secures $> 79\%$ RR, showing recovery supervision is a universal training signal rather than an emergent feature of scale. Third, robustness entails a quantifiable *tradeoff*: retry intensity lowers catastrophic failures (82.4% → 19.3%) while modestly reducing efficiency, situating PALADIN near the Pareto frontier where no baseline dominates on both safety and cost. Fourth, recovery is *retrieval-dependent*: our ablation shows inference-time exemplar matching contributes up to +30 points in robustness. These insights shift the evaluation paradigm. Efficiency alone is a dangerous proxy: our correlation analysis ($r = -0.72$ between CSR and Efficiency) shows that models optimized purely for speed are systematically brittle. Robustness must instead be treated as a first-class objective—on par with accuracy and efficiency—if agents are to be deployed in failure-prone settings.

PALADIN's improvements stem not from scale or external reflection, but from explicit exposure to failures and proactive error handling. The targeted nature of its retries (fewer than one additional step on average) shows resilience does not require brute-force exploration, but data-driven learning of minimal corrective actions.

# 7 CONCLUSION

PALADIN demonstrates that execution-level robustness is not emergent but a *teachable, scalable capability*. By unifying recovery-rich fine-tuning with inference-time exemplar retrieval, it transforms brittle, "happy-path" agents into resilient problem-solvers. Across backbones, PALADIN achieves $> 79\%$ RR, $> 78\%$ TSR, and $> 80\%$ CSR, while keeping efficiency penalties within one step. These contributions extend beyond benchmarks. PALADIN shows recovery behaviors generalize to unseen failures, proving robustness is not tied to specific error types but can be abstracted into transferable policies.

The implications are twofold. For research, robustness supervision provides a new axis for evaluation and learning, complementary to scaling and alignment. For practice, PALADIN enables safer deployment of LLM agents in high-stakes domains where silent failures are unacceptable. Future work should explore adaptive controllers that modulate retry intensity based on task difficulty or model confidence, and integration with error logs from production environments.

In short, PALADIN sets a new benchmark for execution-level resilience: robustness can be systematically taught, generalized across models, and achieved without prohibitive cost—laying the groundwork for safe, failure-aware AI systems.

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

# Appendix

## A    CRITIC-STYLE APPROACH

We implemented a critic-style agent baseline with an oracle-assisted loop as a benchmark for comparison to PALADIN. The critic module (Gemma-12B) was invoked after each tool call to detect execution errors and propose recovery actions. Whenever an error was flagged, the executor was directed to an oracle dataset with pre-defined recovery actions for the detected error type. The most appropriate action among them was then selected to continue the trajectory. A maximum of 3 recovery attempts were permitted per error before execution continued with the latest attempt or failed gracefully.

Our implementation differs from prior CRITIC systems in that, rather than requiring the model to autonomously generate new candidate recovery actions or perform external reasoning (e.g., web search), we provide a structured set of recovery actions from a curated oracle dataset. The model's role was to choose the most appropriate recovery, rather than generate one from scratch.

### EVALUATION EXAMPLE: SERVICEDEPENDENCYFAILURE

**Chosen error:** ServiceDependencyFailure

**Justification:** The request involves two services: fetching hot products from AliExpress and validating email domains. A realistic failure is that one upstream service is temporarily unavailable. The model must recognize this is an external failure, not due to malformed input.

**Expected recovery actions:** Identify the failing service, wait until it is healthy, and retry the request.

## B    TOOLFUZZ

While PALADIN focuses on runtime robustness during tool execution, complementary work such as ToolFuzz addresses pre-execution reliability by improving the alignment between tool documentation and model expectations.

ToolFuzz applies automated fuzz testing to API schemas in order to detect inconsistencies between declared documentation and actual behavior. It uncovered over 20x more specification-related failures than prompt engineering baselines across 32 LangChain and 35 custom tools—revealing widespread documentation underspecification as a root cause of tool-use errors.

Although PALADIN assumes that documentation is accurate at test time (as each tool is provided with correct instructions), ToolFuzz supports our broader vision: robust real-world tool use requires not only runtime adaptability, but also upstream validation of the tool interfaces themselves. We view ToolFuzz as complementary infrastructure—ensuring that PALADIN's recovery logic is exercised on meaningful failures, not avoidable documentation bugs.

## C    RUNTIME ERRORS

| Specific Failure | Example (Simulated Output) | Corrective Action Policy | Rationale and Citations |
|---|---|---|---|
| 400 Bad Request | `"error": "Malformed request syntax", "status": 400` | Re-examine tool documentation, check parameter formatting, and re-issue the call. | The error originates from the client's request. The only path to recovery is for the client (the agent) to correct its own mistake. |
| 401 Unauthorized | `"error": "Invalid API key provided", "status": 401` | Check for a valid API key. If missing or invalid, terminate the task and report the failure. | A non-recoverable authentication error; retrying with invalid credentials is futile. |
| 403 Forbidden | `"error": "User does not have permission for this resource", "status": 403` | Terminate the task and report the lack of permissions. Do not retry. | This is an authorization failure. Retrying will not change permissions. |
| 404 Not Found | `"error": "The requested resource does not exist", "status": 404` | Verify the request URL. If correct, assume the resource is unavailable and try an alternative tool. | A common error caused by typos or moved resources. Check self-error first. |
| 500 Internal Server Error | `"error": "Unexpected server error", "status": 500` | Retry using exponential backoff. If failure persists after 3–4 attempts, terminate and report. | A catch-all for transient server-side issues. Retrying is industry standard. |
| 503 Service Unavailable | `"error": "Service unavailable due to overload or maintenance", "status": 503` | Follow the Retry-After header if present; otherwise, use exponential backoff. | Retry is expected, as the issue is temporary. |
| Request Timeout | `requests.exceptions. Timeout` | Retry with exponential backoff; distinguish connection vs. read timeouts. | Timeouts are usually transient and should be retried. |
| DNS Resolution Error | `requests.exceptions. ConnectionError:geta ddrinfo_failed` | Check hostname for typos. If correct, wait and retry. | Could be a typo or a temporary DNS failure. |
| 429 Too Many Requests | `"error": "Rate limit exceeded", "status": 429` | Respect the Retry-After header or apply exponential backoff. | Standard API rate enforcement. Ignoring leads to blocking. |
| Malformed JSON | `SyntaxError:JSON.par se_error` | Retry first. If repeated, use a lenient parser or fallback tool. | Often due to truncated or corrupted responses. Retry is the simplest fix. |
| JSON Schema Violation | `ValidationError:'abc 'is_not_of_type'numbe r'` | Report a data quality issue. Coerce if allowed, otherwise discard. | Data violates schema expectations. Must be handled gracefully. |

Table 2: Catalog of common failures, simulated outputs, and recovery strategies.

These are just some of the many common errors found in API faults, tool calls, or in daily life. For the rest of the errors used to train this model, check out our github: https://github.com/HexaA2/paladin/tree/main.

# D SYSTEMATICALLY AUGMENTED TRACES

Systematically augmented traces are clean ToolBench style agents–tool execution logs that are enriched with controlled, labeled failures and paired with different recovery trajectories, helping enable training that was reproducible and the evaluation of different recovery behaviors under realistic tool-level faults. This approach helped yield multiple deterministic variants of the same base rollout by injecting specific error types at different steps, with structured annotations for failure detection, diagnosis, and multi-turn correction. Augmentations were generated by applying a failure-policy dictionary over baseline rollouts to inject errors such as timeouts, 5xx responses, rate limits, malformed tool outputs, and partial results, with configurable positions, frequencies, and cascades. Each injected failure was paired with an annotated recovery path (diagnose, replan, parameter fix, retry, or tool-switch), helping yield trajectory-level supervision for LoRA-based SFT and deterministic evaluation via metrics like TSR, RR, CSR, and ES. Because variants are deterministic, they support apples-to-apples ablations (e.g., single- vs multi-fault) and stable comparisons across model sizes and training regimes.

- Base trace: plan → call Tool A with parameters p → receive valid response → continue.
- Augmented variant: identical plan → call Tool A with p → injected timeout at step t → agent detects timeout → backs off with jittered retry and parameter validation → successful response → proceed to completion. A second variant might inject a malformed JSON response instead of a timeout, prompting schema validation, selective re-query, or tool-switch. Each variant includes labels for failure type, step index, recovery rationale, and final outcome, enabling targeted training and evaluation.

Listing 1: Paladin trace with recovery and Finish

```
[{"role": "system","content": [{"type": "text","text": "You are **Paladin
    **, an error-resilient agent ..."}]},
{"role": "user","content": [{"type": "text","text": "I'm conducting a
    research project on social media influencers ..."}]},
{"role": "assistant","content": [{"type": "text","text": "Thought: I need
    to gather user information ...
Action: userinfo_for_instagram_cheapest
Action Input: {
 \"username\": \"nike\"
}"}]},
{"role": "function","content": [{"type": "text","text": "{\"error\":
    \"\", \"response\": \"{'data': {'user': {'biography': 'Spotlighting
    athlete* ..."}]}]},
{"role": "assistant","content": [{"type": "text","text": "Recovery:
    Thought: The API call to retrieve user information ...
Action: user_medias_for_instagram_cheapest
Action Input: {\"user_id\": \"113294420064920\"}"}]},
{"role": "function","content": [{"type": "text","text": "{\"error\":
    \"\", \"response\": \"{\\\"data\\\": [{\\\"media_id\\\":
    \\\"1234567890\\\", \\\"caption\\\": \\\"Just Do It\\\" ..."}]}]},
{"role": "assistant","content": [{"type": "text","text": "Thought: I have
    successfully retrieved the recent media posts ...
Action: Finish
Action Input: {\"return_type\": \"give_answer\", \"final_answer\": \"User
    information and recent media posts for 'nike' ...\"}"}]}]
```

The full corpus, schema, and documentation are publicly released at: https://huggingface.co/datasets/SriVatsa123/Gemma_Ready_For_PALADIN/viewer/default/train?views%5B%5D=train&row=0.

# E  RECOVERY DICTIONARY

At the core of PALADIN's learning process is a recovery dictionary—a curated collection of over 50 execution-level failure types and corresponding recovery strategies. This dictionary was not arbitrarily constructed. Instead, it was built through an extensive review of real-world sources, including:

- Developer forums (e.g., Stack Overflow, GitHub Issues)
- Toolchain documentation (e.g., LangChain, Zapier, API reference manuals)
- Industry engineering blogs detailing agent failure cases (e.g., Google's Bard, OpenAI function calls)
- Academic surveys and benchmarks (e.g., ToolScan, ShieldA, Healer)

We structured our final recovery dictionary around ToolScan's taxonomy of common agentic failures, enriched with patterns synthesized from forums, documentation, and engineering logs. Each error type in the dictionary is paired with recovery actions grounded in real debugging and fallback strategies reported by practitioners.

These error-response mappings were then transformed into example trajectories using the ToolBench format, providing PALADIN with richly annotated failure contexts and realistic correction paths. By rooting this dictionary in actual agent failure logs and developer strategies, we ensured the training data reflected authentic, actionable recovery behavior, not synthetic artifacts.

# F  TOOLSCAN ERROR TYPES

*Tool Hallucination* → tool not found errors

*Argument Hallucination* → missing or invalid tool input

*Invalid Tool Invocation* → tool call failed

*Partial Execution* → incomplete tool outputs requiring continuation

*Output Hallucination* → tool output not valid

*Invalid Intermediate Reasoning* → corrected plans and backtracking in multi-turn dialogues

*Re-entrant Failures* → handled via explicit retry and fallback logic

# G  PYTHON DICTIONARY

Here is part of our dictionary:

Listing 2: Recovery Paths Dictionary

```python
recovery_paths = {

    # Branch 1: Client Request Syntax and Format Errors (400, 422)
    "400_422": [
        {
            "from": "Assistant",
            "value": (
                "Thoughts: The 400 Bad Request and 422 Unprocessable Entity
                    errors indicate client-side issues. 400 "
                "often means malformed syntax or invalid request formatting; 422
                    means the syntax is correct but the "
                "content is semantically invalid.\n\n"
                "Action: Start by examining the request URL and endpoint
                    carefully. Check for typos, extra slashes, or "
                "illegal characters that violate URL standards. Use URL
                    validators or linters. Then, validate your "
                "payload format; if JSON, ensure it is well-formed, properly
                    bracketed, and matches the API schema. Check "
                "required fields and types.\n\n"
                "Verify that all required headers like Content-Type and
                    Authorization are present and correct. For 422, "
                "focus on semantic validation such as correct value ranges,
                    formats, and dependencies."
            )
        },
        {
            "from": "function",
            "value": "Validated URL and headers; payload verified with no
                syntax errors."
        },
        {
            "from": "Assistant",
            "value": (
                "Thoughts: If syntax is good, server-side validation may enforce
                    strict formats or rules not obvious in docs.\n\n"
                "Action: Test with minimal payloads including only required
                    fields; gradually add optional fields to isolate "
                "problematic data. Confirm API version matches docs, and check
                    server logs if available."
            )
        },
        {
            "from": "function",
            "value": "Minimal payload testing performed; error persists."
        },
        {
            "from": "Assistant",
            "value": (
                "Thoughts: Persistent error suggests possible issues with query
                    parameters or headers.\n\n"
                "Action: Verify all query parameters for correctness and allowed
                    values. Confirm proper encoding (UTF-8) "
                "and escaping of special characters. Check multipart boundaries
                    if uploading files."
            )
        },
        {
            "from": "function",
            "value": "Query parameters and encoding verified."
```

```
972        },
973        {
974          "from": "Assistant",
975          "value": (
976            "Thoughts: If no client-side issue found, consider API bugs or
                  server incompatibility.\n\n"
977            "Action: Contact support with detailed logs; implement retry with
978                  exponential backoff."
979          )
980        },
981        {
982          "from": "function",
983          "value": "Escalation logs prepared; retry logic implemented."
984        }
       ],
985
986    # Branch 2: Authentication and Authorization Errors (401, 403, 407)
987    "401_403_407": [
988        {
989          "from": "Assistant",
990          "value": (
991            "Thoughts: 401 Unauthorized indicates missing or invalid
                  credentials, 403 Forbidden means access is denied "
992            "even if authenticated, and 407 Proxy Authentication Required
993                  means the proxy server requires authentication.\n\n"
994            "Action: Check that valid API keys or tokens are sent in headers
                  exactly as required by the service. Refresh "
995            "tokens if expired. Confirm permissions on the user or API key to
                   access requested resources.\n\n"
996            "For 403, verify role-based access controls and resource-specific
997                  permissions. For 407, ensure proxy credentials "
998            "are configured and included properly in requests."
999          )
1000       },
1001       {
1002         "from": "function",
1003         "value": "Credentials and permissions checked; tokens refreshed
                 where needed."
1004       },
1005       {
1006         "from": "Assistant",
1007         "value": (
1008           "Thoughts: Persistent 401/403 might be due to time skew causing
                 token rejection or revoked credentials.\n\n"
1009           "Action: Sync system clock with NTP, regenerate credentials, and
1010                 check for IP restrictions or rate limits "
1011           "that may block access."
1012         )
1013       },
1014       {
1015         "from": "function",
1016         "value": "System time synchronized; no IP blocks detected."
1017       },
1018       {
1019         "from": "Assistant",
1020         "value": (
1021           "Thoughts: If proxy authentication fails, check proxy server logs
                 and credentials.\n\n"
1022           "Action: Update proxy config or switch to a proxy that supports
                 needed authentication methods."
1023         )
1024       },
1025       {
            "from": "function",
            "value": "Proxy credentials updated; connection succeeded."
```

```
1026          }
1027        ],
1028
1029    }
1030
```

The rest of our dictionary can be found in github at https://github.com/HexaA2/paladin.

# H IMPLEMENTATION DETAILS

## H.1 MODEL INSTANTIATIONS

PALADIN was fine-tuned on multiple open-weight backbones to test cross-architecture generality: Gemma-27B, Qwen-2.5-14B-Instruct, AM-Thinking V1, and LLaMA-3.1-8B-Instruct. All models were trained on the full 50K recovery-annotated corpus, enabling direct comparison of robustness transfer across scales and inductive biases. For LLaMA-3.1-8B, extended context via RoPE scaling ensured parity on long recovery traces relative to Gemma and Qwen, avoiding truncation artifacts.

## H.2 LoRA CONFIGURATION

All runs adopted LoRA adapters with rank 16, scaling $\alpha = 32$, and dropout 0.0. Adapters were injected into attention projections (q_proj, k_proj, v_proj, o_proj) and MLP projections (up_proj, down_proj, gate_proj), equipping models with recovery skills while preserving base competence.

## H.3 OPTIMIZER AND SCHEDULER

Training employed paged AdamW (32-bit) with bf16 precision and gradient checkpointing. We used a base learning rate of $2 \times 10^{-4}$, with standard AdamW defaults for warmup and weight decay, and a constant schedule over one epoch, appropriate for single-epoch SFT with LoRA.

## H.4 BATCHING, CONTEXT, AND EPOCHS

Experiments used a context length of 8192 tokens. Training was performed with micro-batch size 1 and gradient accumulation 8 (effective batch size 8), for a single epoch over 50K trajectories (80% failure-rich, 20% clean "happy paths"). This composition balanced recovery supervision with baseline tool-use competence.

## H.5 HARDWARE AND RUNTIME

All fine-tuning ran on h200sxm GPUs. The combination of bf16, paged AdamW, and checkpointing enabled stable 8K-token SFT with LoRA on these accelerators. RoPE scaling extended LLaMA-3.1-8B's effective context to 128K tokens with modest additional memory cost, remaining tractable under h200-class footprints. End-to-end fine-tuning completed within a single epoch per backbone, with wall-clock time increasing monotonically with parameter count (8B < 14B < 27B).

# I   Dataset Construction Details

## I.1   Failure Injection Procedure

We began from ToolBench tasks and tool schemas, discarding original rollouts, and applied an automated trace parser to detect the first execution error. Each trajectory was truncated at that failure point to create a repair target. For every truncated trace, a controller supplied the task, tool schema, error signal, and dialogue context to a GPT-5 Teacher equipped with a recovery dictionary. The Teacher then generated multi-turn `Recovery:` segments consisting of retries, reformulations, fallbacks, or graceful termination, producing repaired trajectories:

$$f_{\text{repair}}(T, A, C, E) \;\; \rightarrow \;\; C',$$

while error-free traces were finalized via:

$$f_{\text{finalize}}(T, A, C) \;\; \rightarrow \;\; C'.$$

The simulator injected failures deterministically by specifying error type, manifestation (e.g., malformed output, silent failure), and turn index. A Python controller executed each scenario by providing tool documentation, applying the designated error, and simulating tool responses to the agent's recovery actions.

**Injected error examples.**

- **Timeouts and 5xx/503:** Transient server failures triggering capped backoff-and-retry before graceful termination.
- **Malformed/invalid outputs:** Truncated JSON, schema violations, or null fields designed to elicit re-queries, lenient parsing, or tool switching.
- **Auth/permission errors (401/403/407):** Non-recoverable or credentials-refresh scenarios; repeated failures prompted terminate-with-explanation policies.

## I.2   Recovery Annotation Process

Recovery supervision combined two sources: (i) a curated recovery dictionary aligned to ToolScan's seven error classes, and (ii) GPT-guided rewriting conditioned on truncated traces and error signals. For each failure, the Teacher expanded dictionary-level strategies into situated multi-turn recoveries in ToolBench format (`Thought → Action → Action Input → Tool Output`), prefixing corrective steps with `Recovery:` tags and concluding with `Finish` plus either a user-facing answer or graceful-failure explanation. Clean "happy path" traces (about 20%) were also audited and, when necessary, rewritten to ensure fully successful interactions, preserving base competence while keeping recovery central in the remaining 80% of traces.

## I.3   Recovery Exemplar Distribution

PALADIN's retrieval bank contains over 55 recovery exemplars derived from the dictionary and aligned to ToolScan's seven canonical error types (see (see Appendix E)). These cover *Tool Hallucination*, *Argument Hallucination*, *Invalid Tool Invocation*, *Partial Execution*, *Output Hallucination*, *Invalid Intermediate Reasoning*, and *Re-entrant Failures*, each paired with exemplar failures and recovery protocols. During execution, observed failures are matched to the closest exemplar via signature distance $d$, with the associated recovery action steering trajectories back to stability. This enables generalization across diverse failure surfaces.

## I.4   Data Split

The final corpus comprises roughly 50K trajectories serialized in ToolBench format with explicit `Recovery:` tags, with an 80/20 composition of recovery-rich to clean traces. Training sequences were processed under a single-epoch CLM SFT regime. Evaluations were conducted in a sandboxed environment with deterministic error injection to ensure controlled, reproducible assessment using TSR, RR, CSR, and Efficiency metrics. For LLaMA-3.1-8B, RoPE scaling extended effective context to 128K tokens, preventing truncation of long recovery trajectories and ensuring parity across backbones under the same split design.

## J PALADINEVAL BENCHMARK

PaladinEval is a deterministic failure-injection benchmark designed to evaluate recovery competence, honesty, and efficiency under the seven ToolScan error classes. It combines controlled simulators, taxonomic labeling, and standardized metrics (TSR, RR, CSR, ES) to enable apples-to-apples comparison across models and methods. Unlike ToolReflectEval, which targets single-call reflective corrections, PaladinEval emphasizes trajectory-level, multi-turn recovery in noisy execution settings. All reported scores are normalized so that higher is better across tables and plots.

### J.1 BENCHMARK DESIGN

PaladinEval instruments ToolBench-style tasks with a simulator that injects a single, labeled execution failure at a specified turn, then drives the episode to completion while logging recovery attempts, retries, tool switches, and termination decisions. Each episode includes the task specification, tool schema, truncated trace at failure, and an injected error drawn from the ToolScan taxonomy. Evaluation is conducted with fixed seeds and deterministic tool outputs to ensure reproducibility across backbones and runs.

### J.2 TASKS AND COVERAGE

The suite spans diverse tool-use tasks representative of training domains and failure surfaces. Episodes are constructed to cover all seven ToolScan error categories under uniform sampling rules, preventing skew toward any particular class. Results reported in the main paper compare PaladinEval against ToolReflectEval across multiple backbones, showing consistent gains in Recovery Rate, Task Success Rate, and Catastrophic Success Rate. These improvements come with expected efficiency trade-offs from retry-heavy strategies, but confirm sufficient breadth and balance across failure classes for comparative evaluation.

### J.3 SAMPLING PROCEDURE

For each task, the first tool failure is injected deterministically by specifying the error class, manifestation (e.g., 5xx timeout, malformed JSON), and turn index. The remainder of the episode is executed with fixed tool responses to recovery actions, minimizing variance. Sampling ensures per-class coverage across *Tool Hallucination*, *Argument Hallucination*, *Invalid Invocation*, *Partial Execution*, *Output Hallucination*, *Invalid Intermediate Reasoning*, and *Re-entrant Failures*, enabling both per-class and macro-averaged reporting of RR, TSR, CSR, and ES.

### J.4 ADAPTING TOOLREFLECTEVAL

For comparability, ToolReflectEval was re-run under the same deterministic simulator and normalized metrics. While ToolReflectEval emphasizes critique-based improvements to single tool calls, PaladinEval stresses multi-turn recovery after explicit failures, capturing behaviors such as diagnosis, replanning, retries, tool switches, and graceful termination. Together, the two benchmarks provide complementary perspectives on robustness.

### J.5 FILTERING AND DEDUPLICATION

Benchmark construction applies truncation at the first failure and removes trajectories with ambiguous or duplicate error signatures to avoid double-counting or conflating error classes. Episodes with inconsistent tool schemas or non-reproducible outputs are excluded to preserve determinism. Clean "happy-path" episodes are retained for competence checks but not scored as recoveries, ensuring evaluation focuses squarely on execution robustness.

# K ADDITIONAL METRICS AND ABLATIONS

## K.1 METRIC VARIANTS AND FORMULAS

We evaluate PALADIN with four metrics:

$$\text{Task Success Rate (TSR)} = \frac{\text{\# successful tasks}}{\text{\# total tasks}},$$

$$\text{Recovery Rate (RR)} = \frac{\text{\# failures recovered}}{\text{\# failures encountered}},$$

$$\text{Catastrophe Success Rate (CSR)} = 1 - \frac{\text{\# hallucinated successes}}{\text{\# total failures}},$$

$$\text{Efficiency Score (ES)} = \frac{1}{\text{average \# steps to complete task}}.$$

RR, CSR, and ES are novel contributions that capture execution-level robustness beyond traditional task success. For diagnostic checks, we also experimented with call-level efficiency and normalized efficiency variants; these preserved model rankings and are omitted from main results for clarity.

## K.2 ABLATION: REMOVING INFERENCE-TIME RETRIEVAL

PALADIN uses taxonomic retrieval over a curated bank of 55+ recovery exemplars aligned to ToolScan to map observed failures to prototypical recovery actions. Removing retrieval disables exemplar matching and forces purely end-to-end behavior. Across backbones, this sharply reduces robustness:

- **Gemma-12B:** RR 89.7% → 61.4%, TSR 87.4% → 57.3%, CSR 82.6% → 65.1%.
- **Qwen-14B:** RR 94.7% → 73.3%, TSR 79.5% → 66.3%, CSR 94.6% → 68.9%.
- **LLaMA-8B:** RR 79.8% → 48.6%, TSR 78.7% → 42.7%, CSR 80.7% → 57.4%.
- **AM-Thinking V1:** RR 96.1% → 81.2%, TSR 81.2% → 70.9%, CSR 88.7% → 73.3%.

Drops of 20–30 points highlight that learned recovery patterns are substantially amplified by exemplar-guided retrieval at inference.

## K.3 ABLATION: TRAINING DATA COMPOSITION

PALADIN is trained on recovery-rich traces (80%) plus clean "happy-path" traces (20%) to preserve tool competence and avoid overfitting to failure-only dynamics. Training solely on injected failures (without Teacher-authored `Recovery:` continuations) degrades multi-turn behavior: agents overfit to "retry-once" heuristics, miss plan-shift and tool-switch transitions, and exhibit decreased CSR due to missing supervised end states. Quantitatively, RR and TSR fall relative to full recovery-annotated training, with larger ES penalties from inefficient repeated retries. This confirms that explicit recovery annotations are essential for learning stable, compositional recovery behaviors.

## K.4 ROBUSTNESS: ZERO-SHOT TRANSFER TO UNSEEN TOOLS

To evaluate generalization, we tested on held-out APIs under the same simulator and error taxonomy. Zero-shot transfer preserves a large fraction of recovery performance, with smaller absolute drops in CSR than RR/TSR. This suggests PALADIN preserves honesty under uncertainty even when recovery is incomplete. Failure analyses show strong transfer of schema-mismatch handling and malformed-output repair, while tool-specific authentication and pagination account for most residual errors. Retrieval mitigates these by guiding toward nearest exemplars, even when the exact tools are unseen.

## K.5 TAKEAWAYS

- Inference-time retrieval is a primary driver of robustness, complementing training-time exposure to diverse failures.

- Recovery-annotated trajectories are critical; failure-only training under-specifies chained recovery, degrading end-to-end completion and safety.
- Zero-shot transfer demonstrates that execution-level recovery behaviors generalize, with retrieval providing a safety net for novel APIs.

## L EXPANDED FIGURES

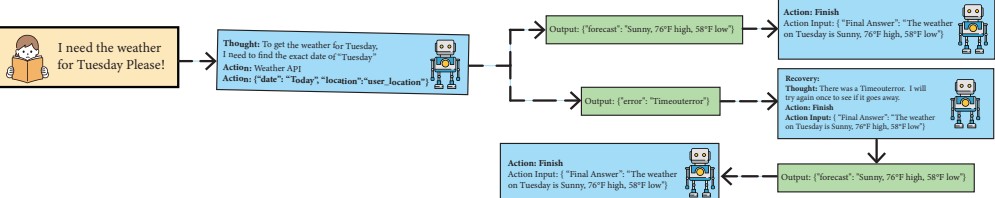

Figure 5: PALADIN's Thought Process

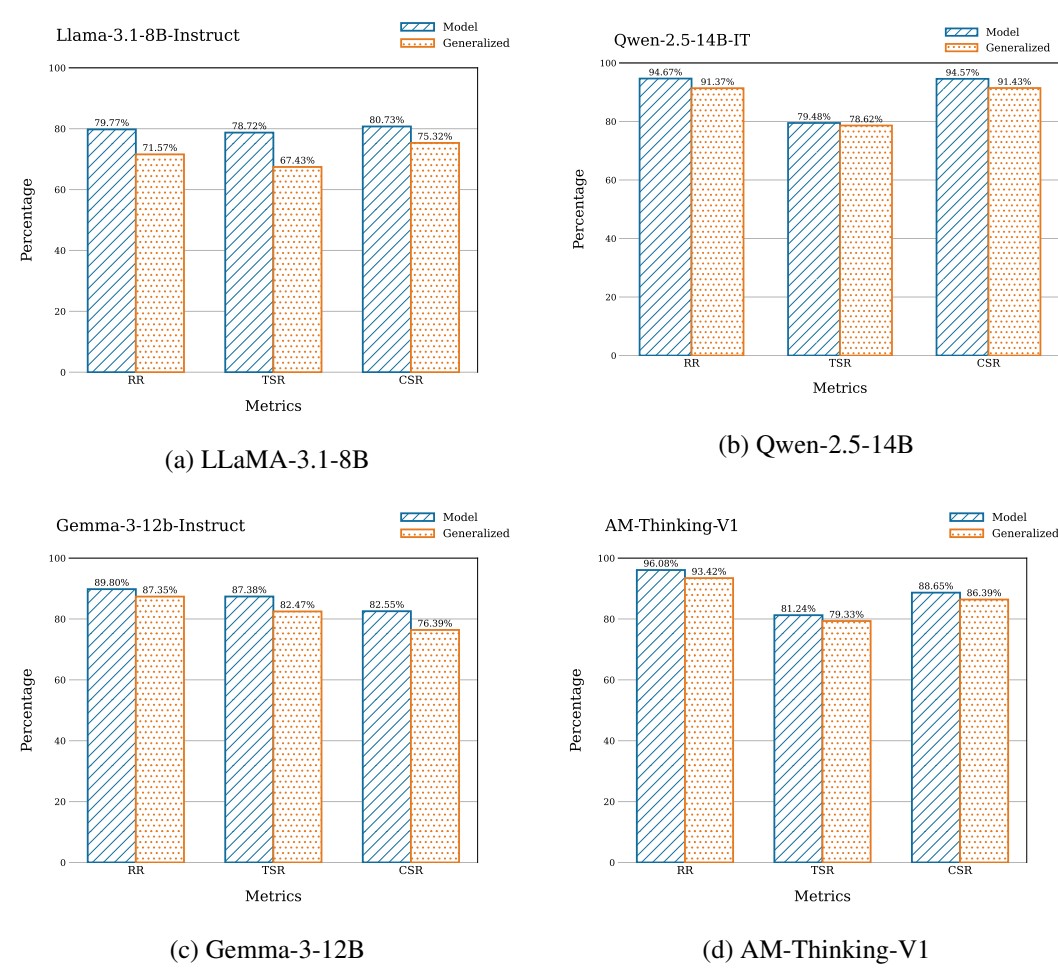

(a) LLaMA-3.1-8B

(b) Qwen-2.5-14B

(c) Gemma-3-12B

(d) AM-Thinking-V1

Figure 6: PALADIN's robustness when facing unseen error types across different model backbones.

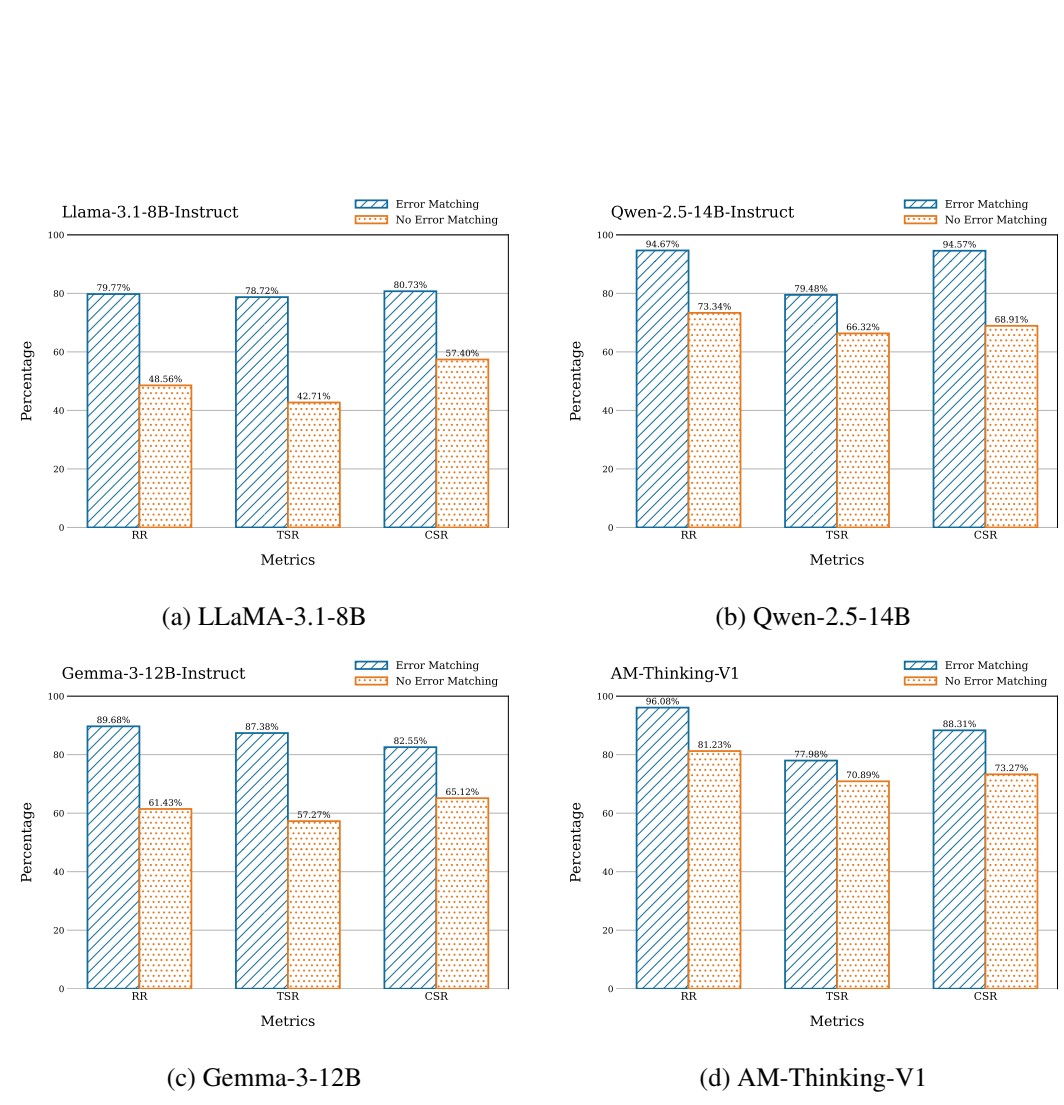

Figure 7: PALADIN's abalation when facing unseen error types across different model backbones.

