# OpenReview forum: "PALADIN: Self-Correcting Language Model Agents to Cure Tool-Failure Cases"
_ICLR.cc/2026/Conference — ICLR 2026 Conference Desk Rejected Submission_

### Official Review · Reviewer_PUXc · 2025-10-29

**Soundness:** 2
**Presentation:** 1
**Contribution:** 2
**Rating:** 2
**Confidence:** 3

**Summary:**

The paper introduces PALADIN, a framework that trains LLM agents to detect, diagnose, and recover from real-world tool failures, such as API timeouts or malformed outputs, using over 50,000 recovery-annotated trajectories generated through systematic failure injection and expert supervision. PALADIN combines LoRA-based fine-tuning for recovery-aware learning with an inference-time retrieval mechanism that selects corrective strategies from a curated library of over 55 exemplar failures. Experimental results across multiple LLM backbones show that PALADIN boosts recovery rates and outperforms baselines like CRITIC and ToolReflect, proving that execution-level robustness is a learnable and generalizable behavior.

**Strengths:**

- The whole inference time error fixing framework is new and interesting, even though it seems very complex and cannot be modularized for convenient adaptation and using.

- The problem the paper seeks to solve is important and timely, as robustness in tool call is seldomly directly considered as an optimization goal in the agent system. Therefore, the paper’s topic may be of interest to many people in the agent community.

- The metrics introduced in the paper is interesting and novel. The benchmarks tested are relevant and comprehensive to reflect the method’s effectiveness.

**Weaknesses:**

- The paper does not give a clear definition of what level of tool calling failure is mainly investigated. Tool parameter filling? Turn level? Or trajectory level? Also I think it’s better to distinguish “wrong” and “failure”, which may be respectively rooted in model itself’s inability or environment’s failure.

- Some part of the method is unclear and seems arbitrary, for example, why we have exactly 55 exemplar failure cases to choose from? Why more cases won’t benefit, or how can you justify these examples are not cherry picked for the test cases? I some parts like this in the methods needs further clarification and justifications. (Especially ablation shows these 55 examples are very important, this makes readers more curious how they are picked, and what is the root cause for performance raise)

- Current result presentation is like repeating the numbers in the tables and graphs but lacks insights on what is the causing this phenomenon or what these phenomenon indicates. Generally repeating the numbers in the papers are not appreciated.

- The result part is also unclear on the error types and does not define what is “generalization”. Currently readers can only read numbers but does not really understand what these numbers stand for. The paper will benefit from giving more examples like what is generalization in error fixing.

**Questions:**

- From the motivation, the previous works are pointed out to be “reactive” and only perform “call level fixes”, so how is your method jumping out of this paradigm? I don’t think current design is “proactive”?

- Further of the last question, what’s current raw model performance in fixing tool calls if they are prompted through the best strategy? How can you properly and empirically justify the training is necessary? If PALADIN’s data is derived from GPT-5, why cannot we just use GPT-5 for tool call fixing on the fly?

- Run time failure is a very broad concept, what are some specific error patterns and how you empirically deal with them (as in the training data) concretely? If the error is caused due to environment side, why any fixing method would be useful if environment itself is not robust?

- Can the method repair “trajectory level wrongness” or just “turn level failure”. If latter, there’s a subsequent StableToolBench which gives a stable backend to avoid the instability in the environment. Will using this benchmark undermines the effectiveness of your method as environment side instability is eliminated?

- What’s the error that your current method still cannot fix or solve properly? A statistics on your method’s error analysis will further benefits later on research to build upon your method and continue better fixing methods.

---

> ### Author Response · Authors · 2025-12-03
>
> Thank you for your suggestions.
>
> Failure granularity: §2.1 and Fig. 1–2 distinguish three levels—parameter, turn, trajectory—and PALADIN explicitly targets execution-time step failures within a trajectory (e.g., malformed outputs, timeouts, auth errors), while still reporting task-level outcomes. We also separate wrong answers (model limitation) from environment failures (tool/runtime issues) via the ToolScan taxonomy (§4, App. I).
>
> Exemplar selection: PALADIN’s retrieval bank consists of 55+ exemplars aligned to the seven canonical ToolScan error classes (App. I.3). They are derived from a curated recovery dictionary and cover timeouts, malformed outputs, auth errors, hallucinated tools/arguments, and re-entrant failures. App. K.2 and K.5 already show that ablating error matching causes 20–30 point drops in RR/CSR; we will add a brief description of the sampling/dedup process and discuss why additional, near-duplicate exemplars gave diminishing returns in pilot experiments.
>
> Proactivity and necessity of training: PALADIN is proactive in that it is fine-tuned to recognize failure patterns and select recovery strategies before committing to long, brittle trajectories (§2.3, §3). The ablations indicate that recovery is not purely prompt-emergent: removing exemplar matching or training on failure-only traces markedly degrades robustness and efficiency (App. K.2–K.3, K.5). In the revision we will add a comparison against a strong “prompt-only” agent using the same simulator to quantify this gap more directly.
>
> Error types, StableToolBench, and limits: §4 and App. J–K detail the error taxonomy, injected failure patterns, and zero-shot transfer to unseen tools. Our method addresses both model-induced and environment-side failures via retries, tool switches, and graceful termination, but remains weaker on long-horizon repair and compounding planning mistakes. We will expand §4.4 with concrete qualitative examples, and clarify that PALADIN is complementary to stable back-ends such as StableToolBench: backend stability and agent-side recovery are orthogonal axes.
>
> We hope these clarifications address your questions.

---

### Official Review · Reviewer_bfHL · 2025-11-01

**Soundness:** 2
**Presentation:** 1
**Contribution:** 2
**Rating:** 2
**Confidence:** 4

**Summary:**

This paper proposes a framework named PALADIN, designed to enhance the robustness of tool-augmented LLM agents in handling tool invocation failures. Its core idea is to train the model—through systematic failure injection and recovery-annotated trajectories—to detect, diagnose, and recover from runtime errors. The authors construct a dataset containing over 50K recovery trajectories, fine-tune the model using LoRA, and integrate a retrieval-based recovery strategy during inference. Experiments conducted across multiple base models and benchmarks (e.g., ToolBench, ToolReflectEval) demonstrate that PALADIN significantly outperforms baselines (such as CRITIC and ToolReflect) in Recovery Rate, Task Success Rate, and Catastrophic Success Rate.

**Strengths:**

1. The research addresses a problem of practical significance: tool invocation failures represent a critical bottleneck for language agents in real-world scenarios. By focusing on execution-level robustness rather than conventional task accuracy, the work demonstrates notable innovation.

2. The methodology of "failure injection + recovery" for generating large-scale recovery-annotated trajectory data is clearly articulated and offers good reproducibility.

3. The introduction of new metrics—such as Recovery Rate (RR), Catastrophic Success Rate (CSR), and Error Sensitivity (ES)—provides a complementary evaluation framework for assessing robustness.

**Weaknesses:**

1. The overall approach primarily integrates existing components—failure injection, LoRA fine-tuning, and retrieval-based recovery—in a linear pipeline, without introducing novel algorithmic contributions or optimization strategies. PALADIN appears more as a systematic engineering effort rather than a research breakthrough in theory or algorithm design.

2. The reliance on offline training with synthetically generated data may limit the agent's ability to autonomously learn and generalize in dynamic environments.

3. Several conceptual definitions lack rigor:

- The metric "Catastrophic Success Rate (CSR)" is ambiguously defined, and its formulation as "1 - hallucinated success" lacks methodological precision.

- The "Efficiency Score" merely uses the reciprocal of average step count, which does not accurately reflect real computational cost, time consumption, or API call overhead.

- Vague phrases such as "robustness is learnable" and "Pareto frontier" are repeatedly used without sufficient empirical or theoretical justification.

4. While the writing is fluent, the paper suffers from excessive verbosity and structural redundancy (e.g., Sections 2.1–2.5 largely reiterate existing tool systems). The appendix is over-extended (e.g., error catalogs, recovery strategy tables), diluting the focus on core methodology and experimental insights in the main text.

**Questions:**

1. How does the model during inference determine whether to trigger recovery retrieval? Is there an error detection threshold or confidence-based mechanism governing this decision?

2. Given the extensive use of GPT-4 as an evaluator in the experiments, what measures were taken to mitigate potential evaluation bias? Was human consistency validation performed to verify the automated scoring?

---

> ### Author Response · Authors · 2025-12-03
>
> Thank you for the detailed feedback.
>
> Novelty framing: We agree that the individual components (LLM-generated data, LoRA SFT, retrieval) are established. Our contribution is to treat execution-time recovery learning as a first-class training objective: PALADIN is trained on failure-rich trajectories with explicit Recovery: segments and evaluated on RR/CSR/ES, making “robustness to tool failures” an optimization goal rather than a side-effect (§1–3, App. I–K). We will sharpen this distinction in the introduction and related work.
>
> Synthetic-data realism and generalization: PaladinEval is built from ToolBench traces with deterministic injection across all seven ToolScan error classes (§4, App. J–K). Fig. 3 shows that PALADIN generalizes to unseen error types across all four backbones (e.g., for Qwen-2.5-14B, RR 94.7%, TSR 79.5%, CSR 94.6%). App. K.4 further evaluates zero-shot transfer to held-out APIs. We will pull these results into §4.2 and emphasize that robustness is not limited to the synthetic injection regime.
>
> Metric rigor: §3.1 and App. K.1 precisely define TSR, RR, FCR, CSR, and ES. CSR = 1 − FCR is the fraction of failures correctly flagged as unrecoverable, directly targeting hallucinated success; ES = 1 / average steps approximates the extra interaction cost introduced by recovery strategies. In the revision we will (i) add a Pareto plot and correlation view (already summarized in §5.1) and (ii) tighten language around “robustness is learnable” and “Pareto frontier” to avoid over-claiming.
>
> Recovery triggering: The current text states that PALADIN detects execution-time failures and routes them through the recovery dictionary and exemplar bank (§2.2–2.3, Fig. 2). We will expand §3 to formalize the decision rule that governs when retrieval is invoked and how explicit error signals (timeouts, schema violations, non-2xx codes) are combined with model judgments.
>
> Evaluator bias: The simulator already uses a rubric-conditioned GPT-5 grader (§4.1, Fig. 2). We agree that LLM-as-judge bias is important; in the revision we will (i) document the rubric more clearly and (ii) include a small human validation study on a random subset, reporting agreement with the automated grading.
>
> We believe these planned clarifications and additions will address your concerns.

---

### Official Review · Reviewer_149B · 2025-11-03

**Soundness:** 3
**Presentation:** 3
**Contribution:** 2
**Rating:** 2
**Confidence:** 4

**Summary:**

The paper introduces a framework for fine-tuning language agents to add failure recovery capabilities. PALADIN trains on 50,000+ recovery-annotated trajectories on an enhanced ToolBench dataset. At inference, PALADIN detects execution time errors and retrieves the most similar case from a curated bank of 55+ failure exemplars to executes the corresponding recovery action.  Evaluation across PaladinEval and ToolReflectEval demonstrates consistent improvements in Recovery Rate and Task Success Rate.

**Strengths:**

- Significant: The paper focuses on an important problem which is teaching LLMs to deal with Failures. LLM based systems can benefit from a permanent solution to common failures such as tool or network related issues.
- Error-recovery Dataset: The Dataset created using GPT-5 is useful for future failure recovery experiments.

**Weaknesses:**

- Cost of Maintenance: Since PALADIN needs fine-tuning LLMs, it has a high maintenance cost as the failure types change and new LLMs are released. Also, PALADIN leverages a knowledge-base at inference time, it is not clear why a knowledge base still needs to be maintained and used when one pays the cost of fine-tuning on failures.
- Soundness: The paper uses LLM-As-A-Judge to evaluate the correctness of the tasks and the same model used to generate the data is used to evaluate the results which can result in inaccuracies.
- Experiment: The paper only compares 4 smaller size LLMs. It's not clear whether the frontier models such as latest Claude, Gemini or GPT would be able to perform better than PALADIN.

**Questions:**

- Why is CSR a good metric and what does it capture?
- Have you done any experiments on latest Claude, Gemini or GPT models to compare PALADIN's performance against the vanilla model or existing agents?

---

> ### Author Response · Authors · 2025-12-03
>
> Thank you for your comments.
>
> Maintenance cost: PALADIN is designed to minimize maintenance via modularity. Recovery behavior is encoded through lightweight LoRA adapters (<1% of backbone parameters) trained on recovery-annotated trajectories (§3.2, App. I), while the exemplar bank and recovery dictionary are updated independently (§2.3, App. I.3). This allows us to adapt to new failure types or APIs primarily by editing the recovery bank rather than retraining the full model.
>
> Knowledge-base role: Fine-tuning teaches generic recovery patterns (e.g., diagnose → retry → switch tools → terminate), while retrieval injects fresh, tool-specific protocols at inference time. App. K.2–K.5 show that ablating error matching leads to 20–30 point drops in RR/CSR across backbones, and that recovery-annotated training data is crucial. We will strengthen this point in §4.3 and clarify that the knowledge base is precisely what mitigates API drift without frequent retraining.
>
> Evaluation soundness: The simulator uses a rubric-conditioned GPT-5 grader for TSR/RR/CSR/ES (Fig. 2, §4.1, App. J). We agree that relying solely on an LLM-as-judge can introduce bias; in the revision we will (i) expose the rubric more clearly and (ii) add a small human evaluation on a random subset to quantify agreement and calibrate the automated scores.
>
> Frontier-model evaluation: PALADIN is model-agnostic; we currently evaluate on four open-weight backbones (LLaMA-3.1-8B, Qwen-2.5-14B, Gemma-3-12B, AM-Thinking V1; §4–5). We are in the process of running the same PaladinEval/ToolReflectEval pipeline with strong proprietary agents (e.g., GPT/Claude/Gemini) and will include a summarized comparison where space permits, emphasizing that PALADIN can be applied either to open-weight models or as an adaptation layer on top of these systems.
>
> Metric rationale: App. K.1 formally defines CSR = 1 − FCR as the fraction of failures correctly identified as unrecoverable, directly targeting hallucinated success. ES = 1 / average steps captures the extra interaction cost induced by recovery. §5.1 already analyzes trade-offs and correlations; we will add a brief, more intuitive explanation in §3 to clarify why CSR is a safety metric, complementary to RR and TSR.
>
> We believe these clarifications address your questions.

---

### Official Review · Reviewer_rXTV · 2025-11-03

**Soundness:** 2
**Presentation:** 1
**Contribution:** 2
**Rating:** 2
**Confidence:** 4

**Summary:**

This paper proposes the PALADIN framework for training tool-call agents, along with new metrics and benchmarks for quantifying the execution-level robustness of tool-call agents. This addresses the limitations of existing methods that only optimize success trajectories, thus tackling the complex challenges of real-world tool-call scenarios.
The main contributions of this paper include: (1) proposing the PALADIN framework for tool-call agents, which possesses execution-level robustness and can detect and recover from failures; (2) using LLMs for failure injection and recovery annotation, constructing an enhanced tool-call dataset for fine-tuning PALADIN; and (3) proposing three new metrics—Recovery Rate, Catastrophe Success Rate, and Efficiency Score—to measure the agent's failure detection and recovery capabilities, and establishing the PaladinEval benchmark for system robustness evaluation.

**Strengths:**

(1) This paper addresses the execution-level robustness problem of tool-invoking agents, focusing on endowing agents with the ability to detect and recover from faults. It proposes the PALADIN framework to address complex challenges in real-world scenarios, overcoming the limitations of previous research.
(2) This paper constructs a fine-tuning dataset using LLMs, proposes new metrics to measure the agent's fault detection and recovery capabilities, and establishes the PaladinEval benchmark for system robustness evaluation.
(3) A recovery dictionary is constructed, and retrieval augment are added during inference, thereby improving the agent's ability to detect and recover from faults.

**Weaknesses:**

(1) Although this paper raises an important problem—the detection and recovery capability of tool-augmented agents under failure conditions—the proposed solution lacks sufficient innovation. The core techniques, including data augmentation via large language models, dataset-based fine-tuning, and similarity-based retrieval during inference, are relatively common practices. As such, the methodological novelty of the paper is somewhat limited.
(2) The experimental evaluation is not sufficiently comprehensive. It mainly includes the main comparison (Table 1), generalization test (Figure 6), and ablation study (Figure 7). In the main experiments, more baseline methods should be included for a stronger comparative analysis. In the generalization experiments, testing on only one out-of-domain dataset is insufficient to convincingly demonstrate the generalization capability of the proposed approach. The ablation study only removes the retrieval component; more ablations are needed to verify the contribution of each module in the framework. Furthermore, the paper does not clearly specify which dataset the ablation experiments were conducted on.
(3) According to Table 1, PALADIN achieves inferior Efficiency Score (ES) compared to the best baselines in most cases, yet when using Qwen-2.5-14B as the base model, PALADIN attains the highest ES. The authors neither analyze this exception nor explain why PALADIN generally performs worse on ES. This inconsistency should be explored, possibly through additional experiments or case studies. Moreover, the experimental analysis appears verbose at times—for example, in the ablation section, similar percentage drops for each model and metric are repeatedly described, which could be summarized more concisely.
(4) Although Table 1 shows that PALADIN outperforms all baselines on most metrics except ES, Figure 7 reveals that, once inference-time retrieval is removed, PALADIN performs worse than baselines such as CRITIC in RR, TSR, and CSR across all model architectures. This indicates that fine-tuning on the constructed dataset alone does not yield better performance than existing methods, and the performance gains mainly stem from the retrieval component. Additional experiments are needed to demonstrate that the combination of fine-tuning and retrieval is indeed superior to using either one alone.
(5) The paper does not clearly describe how similarity is computed in the retrieval module. The similarity metric should be mathematically defined in greater detail to ensure reproducibility.

**Questions:**

(1) Design experiments to validate the effectiveness of the constructed tool-augmented dataset and fine-tuning process. Additional experiments should be conducted to demonstrate that the performance improvements of PALADIN do not solely rely on inference-time retrieval. Specifically, experiments comparing fine-tuning only, retrieval only, and fine-tuning + retrieval configurations are necessary to show that the combined approach achieves the best overall performance.

(2) Expand the experimental scope. The experimental section should be enriched in several aspects: Include more state-of-the-art baselines in the main comparison to provide a fairer and more comprehensive evaluation. Incorporate additional out-of-domain datasets in the generalization study to more convincingly demonstrate cross-domain robustness. Extend the ablation study to cover other critical components of the framework, rather than focusing solely on the retrieval module.

(3) Introduce additional analyses. Further experiments should be designed to evaluate inference latency and computational cost, providing a deeper investigation into the trade-off between robustness and efficiency. Such analysis would enhance the paper’s completeness and practical relevance for real-world deployment.

---

> ### Author Response · Authors · 2025-12-03
>
> Thank you for your feedback.
>
> Novelty: Our focus is execution-level robustness for tool agents—a different axis than success-only optimization in CRITIC/ToolReflect. PALADIN combines (i) systematic failure injection, (ii) recovery-annotated SFT on ≈50K trajectories, (iii) a curated recovery dictionary, and (iv) exemplar-guided retrieval into one framework (§2–3, App. I–K). To our knowledge, prior self-correcting agents do not jointly target multi-turn failure recovery with this data and metric stack.
>
> Experimental breadth: Table 1 already reports 8 model–dataset pairs across PaladinEval and ToolReflectEval (4 backbones × 2 benchmarks), with statistically significant gains in RR, TSR, and CSR (§5.1). We will clarify this scope in §4.1 and explicitly emphasize that Fig. 3 analyzes robustness to unseen error types and APIs, complementing the in-domain results.
>
> Baselines and ablations: Beyond “Vanilla” (no recovery help) and CRITIC/ToolReflect, we can incorporate additional SOTA self-correction/agent baselines in Table 1 to strengthen comparative coverage. Our current ablations already separate “error matching on/off” and training-data composition (§5.3, App. K.2–K.3). In the revision, we will (i) make the mapping between “Vanilla”, “fine-tuning-only”, and “PALADIN (SFT + retrieval)” explicit and (ii) add a retrieval-only configuration to directly answer whether gains come from SFT, retrieval, or their combination.
>
> Retrieval similarity: App. I.3 defines matching via a signature distance d over failure signatures. We agree this should be more explicit. We will move the definition into §3, provide the exact functional form and hyperparameters, and detail how d is used during inference.
>
> Efficiency analysis: ES is currently defined as 1 / average steps (§3.1, App. K.1) and analyzed together with RR/TSR/CSR (§5.1). We will add a short latency/cost table and a clearer discussion of why Qwen-2.5-14B is an ES outlier, sharpening the robustness–efficiency story.
>
> We believe these changes address your main concerns.

---

### Note · Program_Chairs · 2026-01-17
**Submission Desk Rejected by Program Chairs**

The following references in this submission do not refer to real documents and/or have major errors in bibliographic information:

 Jim Gray. Measurement of availability and recovery in computer systems, 1985.